# Genome-Wide Identification of Novel miRNAs and Infection-Related Proteins in *Leishmania major* via Comparative Analysis of the Protozoa, Vectors, and Mammalian Hosts

**DOI:** 10.3390/pathogens14101068

**Published:** 2025-10-21

**Authors:** Tianyi Liu, Jinyang Qian, Yicheng Yan, Xi Zeng, Zhiyuan Yang

**Affiliations:** 1School of Artificial Intelligence, Hangzhou Dianzi University, Hangzhou 310018, China; 2College of Informatics, Huazhong Agricultural University, Wuhan 430070, China

**Keywords:** comparative genomics, drug targets, host-protozoan interaction, *Leishmania major*, miRNA, protein kinases

## Abstract

*Leishmania major* is a unicellular protozoan that causes cutaneous leishmaniasis in mammals and is mainly transmitted by the sand fly *Phlebotomus papatasi*. However, the contribution of microRNAs (miRNAs) and protein-coding genes to its pathogenic mechanisms remains largely unexplored. In this study, we systematically analyzed miRNAs and protein-coding genes in *L. major*, its insect vector, and mammalian hosts. Comparative genomic analysis revealed 2963 conserved proteins shared among the three groups, highlighting a core set of proteins across protozoa, vectors, and hosts. Among mammals, human proteins exhibited the highest homology with *L. major*, while *P. papatasi* displayed the lowest proportion of homologs. Functional annotation of 94 hypothetical proteins identified 27 infection-related proteins, including 24 protein kinases and three tyrosine phosphatases, which may represent novel therapeutic targets. In addition, an EST-based approach identified 29 novel miRNAs in *L. major*. Phylogenetic analysis indicated that these miRNAs diverged into two distinct evolutionary branches, and homology analysis revealed that seven miRNAs were absent in all mammalian species. For example, miR-10117-3p was detected only in nematode *Heligosmoides polygyrus*. Furthermore, miRNA-gene interaction network analysis highlighted four key genes potentially involved in *L. major* infection. Collectively, our findings expand current knowledge of protozoan virulence by identifying novel miRNAs and infection-related proteins and provide promising candidates for future drug development against leishmaniasis.

## 1. Introduction

Leishmania major is a type of protozoan that causes cutaneous leishmaniasis, a vector-borne disease affecting mammals. Infections with *L. major* typically manifest as localized cutaneous lesions or ulcers at the site of the sand fly bite, characterized by swelling, inflammation, and tissue destruction [1]. Although extensively studied, important knowledge gaps remain, particularly regarding its pathogenic mechanisms and the development of effective interventions. Current control strategies mainly focus on reducing sand fly populations through vector control and promoting community education to minimize exposure [2]. However, no effective vaccine is yet available [3]. Thus, a deeper understanding of the biology and pathogenesis of *L. major* is essential for developing novel therapies, and protozoan control will continue to be a key approach for managing leishmaniasis in the future.

Genome annotation of *L. major* and the studies of its microRNAs (miRNAs) have become particularly important. The first annotated genome of *L. major* was published in 2005 [4] and updated in 2021 [5]. Despite these updates, the functions of many proteins remain unknown, underscoring the need for systematic re-annotation. Advances in bioinformatics have provided powerful tools for protein function prediction, which have been successfully applied to other pathogens such as bacterium *Mycobacterium tuberculosis* [6] and fungus *Alternaria alternata* [7]. These approaches hold promise for improving the annotation of *L. major* proteins.

In addition, relatively few studies have investigated the role of miRNAs in leishmaniasis. miRNAs are small non-coding RNAs, typically 18–24 nucleotides in length, that regulate protein expression at the post-transcriptional level. Evidence suggests that specific miRNAs play important roles in *L. major* infection, such as miR-146a [8]. With advances in computational approaches, novel miRNAs have been successfully identified from expressed sequence tags (ESTs) in species including plant *Jatropha curcas* [9] and protozoan *Trypanosoma brucei* [10].

The urgent need for novel therapeutic strategies against leishmaniasis is highlighted by the limitations of current treatments and the absence of a human vaccine. Recent approaches include combination therapies for cutaneous and visceral leishmaniases; drug repurposing strategies, such as with artesunate; and exploration of natural products with potential anti-parasitic activity. Protein kinases, in particular, have emerged as critical molecular targets for drug discovery in Leishmania, supported by genome sequencing data that provide insights into essential pathways for protozoan survival [11].

In this study, we applied a series of bioinformatics tools to identify protein-coding genes and miRNAs in *L. major*. We functionally annotated protozoan proteins with an emphasis on those related to virulence, and explored specific miRNAs potentially involved in infection. Collectively, our work provides a comprehensive overview of proteins, genes, and miRNAs in *L. major*, offering new insights into its pathogenic mechanisms and potential therapeutic targets. Given the clinical significance of leishmaniasis as a neglected tropical disease, our findings contribute to a better understanding of host–pathogen interactions.

## 2. Materials and Methods

### 2.1. Genome and Proteome Retrieval

The genome of *Leishmania major*, including 8537 protein-coding sequences and expressed sequence tag (EST) data, was retrieved from the National Center for Biotechnology Information (https://www.ncbi.nlm.nih.gov/, NCBI). Proteomes of eight representative mammals and ten insects were obtained from the UniProt database (https://www.uniprot.org/) [12], while miRNA sequences for all animals were retrieved from the miRBase database (https://www.mirbase.org/) [13]. The overall workflow of our study is illustrated in Figure 1.

### 2.2. Protein Homology Analysis

Since *L. major* is transmitted by insects and infects mammalian hosts, we analyzed protein homology across three groups: the protozoa, mammals, and insects. Proteins from eight common mammalian species, including *Bos taurus* (bovine), *Cavia porcellus* (guinea pig), *Cricetulus griseus* (Chinese hamster), *Homo sapiens* (human), *Mus musculus* (mouse), *Ornithorhynchus anatinus* (platypus), *Oryctolagus cuniculus* (rabbit), and *Rattus norvegicus* (rat), were selected for comparison with *L. major* using BLAST (version 2.16) with an E-value cutoff of 1 × 10^−6^. This stringent threshold reduces the likelihood of false positives and thus increases the reliability of homologous alignments.

Similarly, ten insect species, including *Aedes aegypti* (yellow fever mosquito), *Anopheles gambiae* (African malaria mosquito), *Apis mellifera* (honey bee), *Culex pipiens* (common house mosquito), *Drosophila melanogaster* (fruit fly), *Glossina morsitans* (tsetse fly), *Lutzomyia longipalpis* (sand fly), *Musca domestica* (house fly), *Phlebotomus papatasi* (sand fly), and *Pieris brassicae* (large white butterfly), were compared against *L. major* proteins. The distribution of homologous proteins across the three groups was calculated and analyzed.

To assess uncertainty, we calculated standard errors and 95% confidence intervals (CI) as follows:$$SEM=\frac{SD}{\sqrt{N}}$$$$CI=\bar{x}\pm t\;*\;SEM$$

Here, SEM represents the standard error of the mean, SD is the sample standard deviation, N is the sample size, $\bar{x}$ is the sample mean, and t is the t-value corresponding to a 95% confidence interval (α = 0.05) with N − 1 degrees of freedom.

### 2.3. Protein Function Enrichment Analysis

After removing duplicate alignments, we integrated the results and applied a bi-directional best hit (BBH) strategy [14] to identify the intersection set of proteins present in all mammalian species. Gene Ontology (GO) enrichment analysis was performed using the Database for Annotation, Visualization and Integrated Discovery (https://davidbioinformatics.nih.gov/, DAVID) [15]. For each GO term, the number of associated proteins was counted, and terms with a *p*-value ≤ 0.05 were considered significantly enriched. Three GO categories—cellular component, molecular function, and biological process—were analyzed separately.

### 2.4. Hypothetical Protein Annotation

Due to incomplete annotation of the *L. major* genome, many proteins remain labeled as “hypothetical protein.” To improve annotation, these sequences were aligned against the non-redundant (nr) protein database in NCBI using BLAST. The top hits were extracted and used to assign putative functions to the hypothetical proteins in *L. major*.

### 2.5. Infection-Related Protein Identification

Protozoa rely on specific proteins to establish infection in mammalian hosts, making these proteins potential targets for therapeutic intervention. In *L. major*, key protein groups implicated in infection include tyrosine phosphatases, cytokines, and protein kinases. For example, *L. major* can secrete tyrosine phosphatases via exosomes to modulate host cell signaling [16], while cytokine gene expression can alter the function of infected human macrophages [17]. Protein kinases serve as critical regulators in the protozoan life cycle and are considered important molecular targets [18]. Based on these findings, we focused on these three protein groups in our analysis and discussion.

### 2.6. Identification of Novel miRNA

miRNA-mediated approaches have shown potential for future therapeutic applications in various diseases. Given the high conservation of miRNAs across species, novel miRNAs can be identified in target organisms through homology-based searches. Expressed sequence tags (ESTs) are short DNA fragments derived from complementary DNA, which can serve as markers for gene locations. The EST-based method has been widely applied for miRNA discovery in parasites [10] and was employed in this study. All known animal miRNAs were aligned with *L. major* EST sequences using BLAST (version 2.16). Candidate miRNAs were selected based on the following criteria:(1)Maximum of 2 mismatches allowed;(2)Prediction of the minimum free energy frequency (FMFE) of matching sequence fragments using RNAfold (version 2.3) [19], with a cutoff FMFE ≤ 0.05;(3)Prediction of hairpin-like secondary structures using RNAfold (version 2.3);(4)Significant negative values of minimum folding free energy (MFE) and MFE index (MFEI) for secondary structures, calculated using:$$MFEI=\left[\frac{MFE}{Length\;of\;mature\;miRNA}\;*\;100\right]/(G+C)\%$$

These thresholds are consistent with prior studies on RNA structure prediction, ensuring the reliability of our results.

### 2.7. Construction of miRNA Phylogenetic Tree

Phylogenetic analysis of miRNAs provides insights into their evolutionary relationships, helping to identify common ancestral miRNAs, infer potential shared functions, and determine conserved miRNAs across species. Multiple sequence alignment of the identified miRNAs was performed using Clustal Omega (version 1.2.2) [20] to detect common features. Based on the alignment, a phylogenetic tree was constructed using the maximum parsimony method, which seeks the simplest tree minimizing sequence changes. The reliability and stability of the tree were evaluated using a bootstrap-like self-unfolding approach, in which the tree was reconstructed multiple times and node support values were calculated. Nodes with higher support values were selected to construct the final phylogenetic tree, illustrating the evolutionary relationships among miRNAs.

### 2.8. Distribution of Identified miRNAs

Analyzing the distribution of homologous miRNAs can provide insights into their potential functions. The miRNAs identified in *L. major* were compared with homologous sequences in other species, and their distribution patterns were visualized. Given that mammals are the primary hosts of *L. major*, all known mammalian miRNAs were included in the comparison. miRNAs unique to *L. major* were selected for further functional analysis.

### 2.9. Construction of miRNA-Gene Interaction Network

Target genes of the identified miRNAs were predicted using miRWalk [21] and miRTarBase [22]. Based on these predictions, a miRNA–gene interaction network was constructed and visualized using Cytoscape (version 3.9.1) [23]. To identify key functional nodes within the network, topological analyses were performed to highlight critical miRNAs and their associated target genes. The potential roles of these key miRNAs in *L. major* infection were further explored and discussed.

## 3. Results

### 3.1. Protein Homolog Among L. major, Insects, and Mammals

After performing protein sequence alignments, duplicate homologs between *L. major* and eight mammalian species were removed. Among mammals, human proteins shared the highest homology with *L. major*, with 3785 homologous proteins identified. The lowest number of homologs was observed in *O. cuniculus* (rabbit), with 3465 proteins. Overall, the proportion of homologous proteins between *L. major* and the mammalian proteomes ranged from 40% to 45% (Figure 2A). Understanding these differences between pathogen and host may inform the development of drugs that specifically target the protozoan while minimizing host toxicity.

Comparisons with ten insect species revealed that most homologous protein percentages were approximately 40% (Figure 2B), e.g., *D. melanogaster* (39.94%) and *A. gambiae* (40.27%). Notably, *P. papatasi*, the primary vector of *L. major*, exhibited the lowest proportion of homologs with this protozoan. These results indicate that the majority of *L. major* proteins are significantly distinct from those of mammals and insects.

A two-sample *t*-test was conducted using Origin software (version 10.1) to evaluate whether mean homologous protein counts differed significantly between mammals and insects. The null hypothesis assumed no difference, while the alternative hypothesis assumed a significant difference. The results indicated *p*-values ≤ 0.01 (Figure 2C), suggesting a statistically significant difference in homologous protein quantities between mammalian and insect groups. For the mammalian group, the standard error of the mean (SEM) was 36.68 and the 95% confidence interval (CI) was [3627, 3800.5]; for insects, SEM was 38.7 and CI was [3287.46, 3462.54]. Although overall homology proportions are relatively low, further functional analyses may elucidate the roles of these proteins in protozoan survival, host adaptation, and disease progression, providing potential targets for therapeutic development.

Based on sequence alignments, the occurrence frequencies of 4149 homologous proteins across all 18 species (8 mammals and 10 insects) were calculated and visualized (Figure 2D). The results revealed notable differences in protein distribution between species: 3813 proteins were present in at least six species, whereas 2459 proteins were conserved across all 18 species. This distribution provides a foundation for investigating potential host–protozoan interactions and may guide future studies into the molecular mechanisms of *L. major* infection.

### 3.2. Protein Distribution Among L. major, P. papatasi, and Humans

Following BLAST sequence alignments, we conducted a detailed analysis of homolog distribution in *P. papatasi* and humans, representing the insect vector and mammalian host, respectively. Among insect homologs, 13 proteins were absent in all mammals, and 34 proteins were absent specifically in humans (Figure 3A). For *P. papatasi*, 46 proteins were not found in mammals, while 93 proteins were absent in humans. Conversely, among mammalian homologs, 100 proteins were absent in insects and 497 proteins were absent in *P. papatasi* (Figure 3B). These findings suggest that *L. major* proteins exhibit differential similarities across hosts, potentially reflecting variations in host immune responses to protozoan infection.

A Venn diagram illustrating protein homolog distribution across *L. major*, Homo sapiens, and *P. papatasi* revealed that 2963 proteins were shared among all three species, while 4659 proteins were unique to *L. major* (Figure 3C). The number of homologous proteins shared with humans was notably higher than with *P. papatasi*, which may indicate greater adaptability of *L. major* to mammalian hosts. These results provide insights into future studies on the complex mechanisms underlying *L. major* virulence in humans.

### 3.3. Function Enrichment Analysis of Homologous Proteins

Sequence alignments between the proteomes of eight mammals and *L. major* were performed, and the intersection set of proteins present in all mammals was identified using the bi-directional best hit strategy. Gene Ontology (GO) enrichment analysis of these conserved proteins was conducted using DAVID, with a significance threshold of *p* ≤ 0.05. GO terms were categorized into three classes—biological process, cellular component, and molecular function—and visualized using pie charts (Figure 4).

In the biological process category, “mRNA splicing via spliceosome (GO:0000398)” accounted for the largest proportion (17.6%). This process is a critical step in regulating gene expression via processing precursor mRNAs into mature transcripts, thereby influencing protein synthesis and function [24]. Within the cellular component category, “cytosol (GO:0005829)” represented the highest proportion (24.2%), suggesting that many *L. major* proteins function within host macrophage-like cells during infection [25]. In the molecular function category, “protein binding (GO:0005515)” dominated (78%), highlighting the central role of these proteins in mediating protein–protein interactions and potentially participating in key biological processes such as signal transduction and metabolic regulation [26].

These GO enrichment results provide a comprehensive overview of the functional distribution of *L. major* and mammalian homologous proteins. They offer insights into the molecular mechanisms underlying protozoan virulence and host interactions and help to identify functional differences that may contribute to disease development.

### 3.4. Annotation of Hypothetical Proteins

A total of 2346 hypothetical proteins of *L. major* were retrieved from the NCBI database in February 2025. Novel annotations were assigned based on sequence homology to the best-matching proteins identified through BLAST alignment. Using this approach, 94 hypothetical proteins were successfully annotated (Appendix A), with the 15 proteins exhibiting the lowest E-values highlighted in Table 1.

For example, protein XP_001686672.1, previously annotated as “conserved hypothetical protein,” was re-annotated as “Intraflagellar Transport Protein 122” with an extremely low alignment E-value of 2.27 × 10^−149^. In UniProt, this protein remains listed as hypothetical, whereas TriTrypDB annotates it as intraflagellar transport protein 122. Similarly, protein XP_888594.1 was annotated as “Tetratricopeptide Repeat Domain 21B” with an E-value of 1.57 × 10^−117^.

These results demonstrate that homology-based annotation can provide reliable functional insights into previously uncharacterized proteins. Further experimental validation (“wet lab” studies) is warranted to confirm these annotations. Overall, this analysis enhances our understanding of potential mechanisms underlying host adaptation and infection by *L. major*.

### 3.5. Infection-Related Protein Analysis

Previous studies have demonstrated that *L. major* relies on key proteins, including tyrosine phosphatases, cytokines, and kinases, to infect mammalian hosts [16,17,18]. Based on our protein homology analysis, proteins annotated with keywords such as “tyrosine phosphatase,” “cytokine,” or “kinase” were extracted, and detailed results are presented in Table 2. No novel cytokines were identified in *L. major* through homology searches. However, two infection-related protein groups—kinases and tyrosine phosphatases—were newly annotated. In total, 24 protein kinases and three tyrosine phosphatases were identified within the *L. major* proteome. Given their critical roles in signal transduction, these proteins may play essential functions during protozoan infection. For instance, four adenylate kinases (ADK2, ADK7, ADK8, ADK9) were newly identified. Adenylate kinase is a phosphotransferase enzyme that catalyzes the interconversion of adenine nucleotides, suggesting that *L. major* may depend heavily on cellular energy homeostasis to establish infection [27]. These infection-related proteins represent potential drug targets. Inhibition or functional disruption of these kinases and phosphatases could interfere with protozoan growth and virulence, providing avenues for the development of novel therapeutic interventions.

### 3.6. Novel miRNA Analysis

The miRNAs are conserved non-coding RNAs that play crucial roles in post-transcriptional regulation. Using an EST-based approach, which has proven effective for identifying novel miRNAs in published genomes, we compared *L. major* sequences against known miRNAs in the miRBase database and identified a total of 29 miRNAs in the *L. major* genome. The characteristics of these miRNAs, including length, EST accession, and mismatch values, are summarized in Table 3. The miRNA lengths ranged from 18 to 22 nucleotides, consistent with typical miRNA sizes. Mismatch values are critical for distinguishing miRNAs from other non-coding RNAs. Two miRNAs, lma-miR-9984 and lma-miR-146a, exhibited zero mismatches, indicating complete identity with known miRNAs. Eight miRNAs differed by only a single nucleotide (mismatch = 1) from their known counterparts.

Screening miRNAs using a minimum free energy frequency (FMFE) threshold of ≤0.05 revealed that most predicted MFEI values were significantly negative, supporting the stability of these sequences. Furthermore, the majority of miRNAs exhibited well-formed secondary hairpin structures, providing additional evidence of their reliability and potential functionality.

### 3.7. Analysis of miRNA Phylogenetic Tree

To investigate the evolutionary relationships among the 29 identified miRNAs, a phylogenetic tree was constructed (Figure 5). In this tree, miRNAs clustered within the same branch are inferred to share a common ancestor, while branch lengths and node positions reflect the extent of evolutionary changes. The phylogenetic tree can be clearly divided into two main branches: the first branch ranges from lma-miR-4079-3p to lma-miR-4272, and the second branch ranges from lma-miR-2227 to lma-miR-1237. Highly conserved miRNAs often cluster within sub-branches; for instance, lma-miR-3578 to lma-miR-9984 form a sub-branch, suggesting they may retain similar functions across species.

Support values indicated on the tree represent the frequency with which a node or branch appears across multiple reconstructions. Notably, two miRNAs, lma-miR-7008-3p and lma-miR-6886-3p, exhibited the highest support values of 67, indicating that these nodes are highly reliable in representing *L. major* miRNA genome evolution.

### 3.8. Distribution of Identified miRNAs in Mammals

The identified *L. major* miRNAs were compared with homologous sequences in the miRBase database to examine their distribution across mammalian and non-mammalian species (Table 4). Seven miRNAs were found to be present only in non-mammalian species; for example, miR-10117-3p was detected exclusively in nematode *H. polygyrus*, suggesting a potential functional similarity in host-protozoan interactions between *L. major* and *H. polygyrus*. Conversely, miR-3960 and miR-150-3p were present in both *H. sapiens* and *M. musculus*, indicating that these miRNAs may perform conserved functions in mammals. Additionally, miR-146a was detected in four mammalian species (*B. taurus*, *P. troglodytes*, *E. caballus*, and *C. jacchus*), highlighting its potential as a highly conserved and functionally important miRNA. These findings provide a foundation for further investigation into the roles of these miRNAs in host-protozoan interactions and the pathogenic mechanisms of *L. major*.

### 3.9. Analysis of miRNA-Gene Interaction Network

To further investigate the regulatory roles of *L. major* miRNAs, we identified miRNA-targeted genes and constructed a miRNA-gene interaction network using Cytoscape (Figure 6). In this network, six key miRNAs were found to interact with 23 proteins. Notably, lma-miR-142-5p exhibited the highest number of target genes, suggesting its role as a central regulator potentially involved in multiple biological processes. Several proteins in the network represent critical nodes. For example, superoxide dismutase 2 (SOD2) is essential for maintaining mitochondrial integrity and function, as mitochondria are the primary sites of reactive oxygen species (ROS) production during cellular respiration. SOD2 thus protects cells from oxidative damage, regulates apoptosis, and contributes to overall cellular homeostasis [28]. HSPA8, a member of the heat shock protein 70 (Hsp70) family, functions as a molecular chaperone involved in protein folding, stress response, and signal transduction, playing a crucial role in maintaining protein homeostasis and cell survival under various conditions [29]. These findings emphasize the importance of studying the 23 proteins within the network to elucidate their potential roles in the infection mechanisms of *L. major*.

## 4. Discussion

According to World Health Organization (WHO), leishmaniasis affects populations in endemic areas, with an estimated 700,000–1,000,000 new cases and approximately 20,000–30,000 deaths per year worldwide [30]. The current *L. major* genome comprises 8,537 proteins, of which 2319 have unknown functions. To explore the potential roles of these proteins, we performed sequence alignments between *L. major* and the proteomes of eight mammalian species. *H. sapiens* exhibited the highest level of homology, with 3,785 proteins shared. Overall, homologous proteins accounted for 40–45% of the *L. major* proteome across these mammals. A two-sample *t*-test revealed a highly significant difference (*p* ≤ 0.01) in the number of homologous proteins between mammals and insects. Based on these results, we successfully annotated 116 hypothetical proteins, which laid the foundation for subsequent functional analyses. Functional enrichment analysis revealed that “mRNA splicing via spliceosome (GO:0000398)” represented 17.6% of the biological process category, whereas “protein binding (GO:0005515)” accounted for 78% in the molecular function category, indicating a key role in mediating protein–protein interactions.

By aligning against the NCBI non-redundant database, we further annotated 94 hypothetical proteins. Among these, 27 infection-related proteins were identified, including 24 protein kinases and three tyrosine phosphatases. Eukaryotic protein kinases have been reported as promising targets for antileishmanial drug development due to their crucial roles in signal transduction via phosphorylation. Inhibiting these kinases is expected to disrupt essential processes such as cell-cycle progression, differentiation, and virulence, ultimately affecting Leishmania viability and parasitic survival.

While protein kinases and miRNAs emerged as promising therapeutic candidates, the issue of target specificity warrants careful consideration. Because many eukaryotic kinases are evolutionarily conserved [31], cross-reactivity with host kinases remains a potential risk. However, several of the kinases identified here show Leishmania-specific sequence motifs and domain architectures distinct from their mammalian counterparts. For instance, multiple kinase domains display insertions, deletions, or divergent activation-loop residues unique to trypanosomatids, suggesting that selective inhibition may be achievable. Future structural modeling and comparative docking analyses could help pinpoint parasite-specific pockets that differ from those in host homologs. Similarly, transcriptomic and proteomic profiling during infection can further verify parasite-specific expression, minimizing off-target effects on host cells. Collectively, integrating sequence, structural, and expression data will be critical to prioritize Leishmania-specific targets and enhance the translational relevance of these findings.

In addition, 29 miRNAs were identified in the *L. major* genome by comparison with known sequences in miRBase. Phylogenetic analysis divided these miRNAs into two distinct branches: the first from lma-miR-4079-3p to lma-miR-4272, and the second from lma-miR-2227 to lma-miR-1237. Homology analysis revealed that seven miRNAs (lma-miR-10117-3p, lma-miR-2227, lma-miR-2g-5p, lma-miR-10849, lma-miR-4171-5p, lma-miR-4079-3p, and lma-miR-4057-3p) were absent in all mammals, suggesting their potential as protozoa-specific regulatory molecules and targets for drug development.

The miRNA-gene interaction network further highlighted key regulatory genes, including MAPK1 and GSK3B. MAPK1 (Mitogen-Activated Protein Kinase 1) plays a central role in signal transduction, transmitting extracellular stimuli from the cell surface to the nucleus [32,33]. GSK3B (Glycogen Synthase Kinase 3 Beta) is a serine/threonine kinase involved in diverse cellular processes, including autophagy and glycogen metabolism. It phosphorylates and inhibits glycogen synthase, thereby regulating glucose storage [34,35]. These findings underscore the presence of critical genes and pathways associated with *L. major* infection and provide a foundation for future studies on host–parasite interactions and therapeutic interventions.

Importantly, our integrated analysis of protein-coding genes and miRNAs provides a comprehensive view of the molecular machinery underlying *L. major* virulence. By combining functional annotation, enrichment analysis, and miRNA-gene network construction, we identified protozoan-specific molecules and infection-related pathways that may serve as novel drug targets or biomarkers. This study not only enhances our understanding of the molecular basis of *L. major* infection but also establishes a framework for future experimental validation and therapeutic development.

Our study offers valuable insights into the complex interplay among *L. major*, its insect vector, and mammalian hosts. The identification of novel miRNAs and infection-related proteins opens new avenues for antileishmanial drug discovery, while the functional and evolutionary analyses provide a foundation for understanding protozoan adaptation and host specificity. Future research should focus on experimental validation of these targets, the mechanistic roles of identified miRNAs, and the development of targeted interventions to mitigate the global burden of cutaneous leishmaniasis.

In addition to parasite-derived molecular targets, vector–parasite interactions represent a critical but often underexplored aspect of leishmaniasis transmission. *P. papatasi*, the principal vector of *L. major*, exhibits species-specific molecular determinants, such as midgut receptors and salivary proteins, that facilitate parasite attachment and survival [36]. Comparative analyses of vector transcriptomes and proteomes have revealed unique signatures that could serve as biomarkers for potential targets to interrupt transmission. Integrating these vector-specific findings into therapeutic and diagnostic frameworks, such as designing inhibitors that disrupt parasite-vector binding, may enhance the translational potential of *Leishmania* research. Future work should thus aim to combine parasite and vector omics data to better understand and control leishmaniasis at the molecular interface between parasite and vector.

To experimentally validate our computational predictions, several complementary strategies can be employed. First, the parasite-specific miRNAs identified in this study could be functionally characterized using gain- and loss-of-function assays, followed by quantitative assessment of parasite growth and infectivity. The downstream target genes predicted from the miRNA-gene interaction network could be verified through dual-luciferase reporter assays and RT-PCR analyses to confirm regulatory relationships [37]. Second, kinase inhibition studies using specific small-molecule inhibitors could help determine the functional importance of the identified infection-related kinases in protozoan viability and virulence. Third, macrophage infection models could be applied to evaluate how modulation of these miRNAs or kinase genes affects host–parasite interactions, cytokine responses, and intracellular parasite survival. Together, these approaches will provide direct evidence supporting the bioinformatic predictions and strengthen the mechanistic understanding of *L. major* virulence.

## 5. Conclusions

We identified novel protein-coding genes and miRNAs in *Leishmania major* through comparative genome analysis with its insect vector and mammalian hosts. Several infection-related proteins and miRNAs were highlighted as potential biomarkers and drug targets. These findings advance our understanding of *L. major* virulence and provide a foundation for future antileishmanial therapy development.

## Figures and Tables

**Figure 1 pathogens-14-01068-f001:**
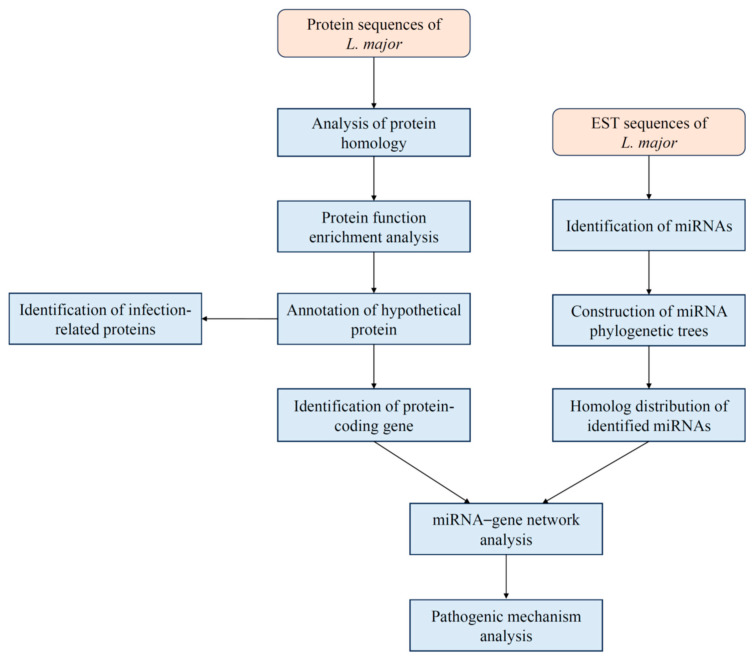
Analytic flowchart of protein-coding genes and EST sequences of *L. major* in this study.

**Figure 2 pathogens-14-01068-f002:**
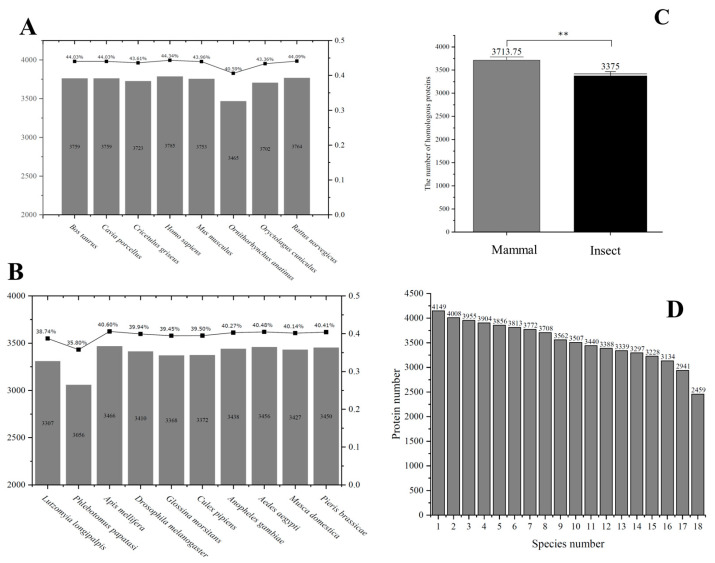
Protein homologs between *L. major*, insects, and mammals. (**A**) The homolog number of *L. major* when compared with mammals; (**B**) the homolog number of *L. major* when compared with insects; (**C**) statistical analysis of homolog number between these two groups; (**D**) frequency of occurrence of *L. major* in the distribution of homologs in 18 species. Two consecutive asterisks (**) indicated *p*-values ≤ 0.01 in statistics.

**Figure 3 pathogens-14-01068-f003:**
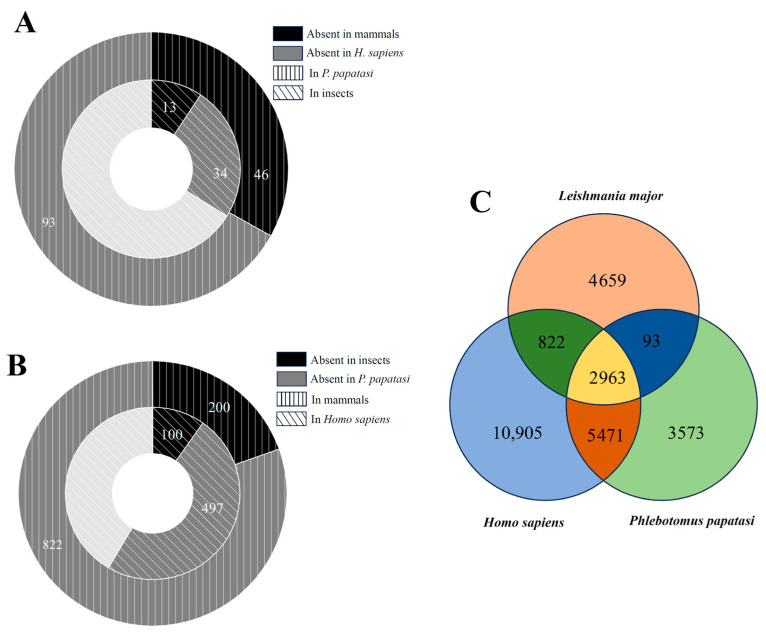
Protein homology distribution among *L. major*, *P. papatasi*, and *H. sapiens*. (**A**) Proteins absent in *H. sapiens*; (**B**) proteins absent in *P. papatasi*; (**C**) Venn diagram of three species.

**Figure 4 pathogens-14-01068-f004:**
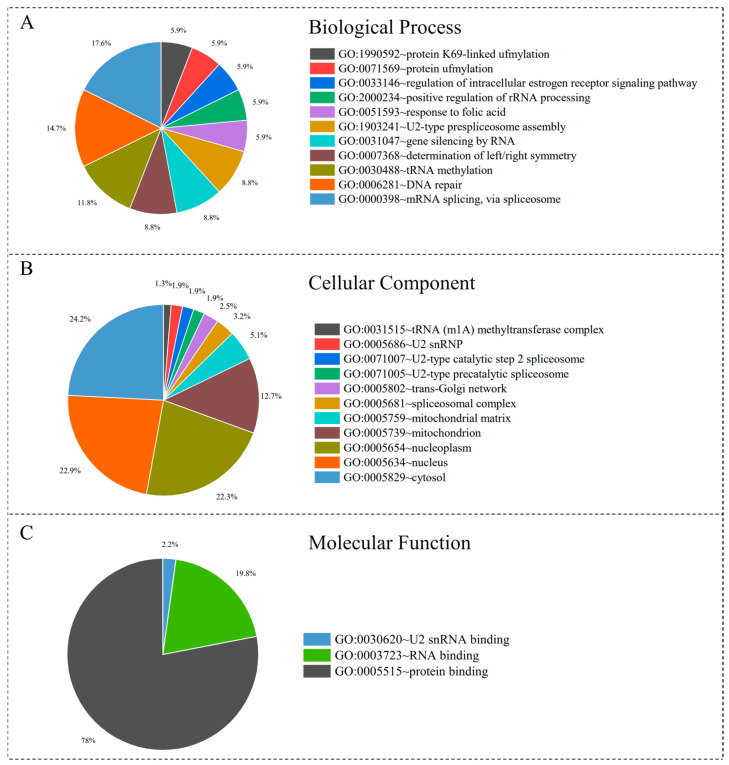
Gene ontology enrichment analysis of homologous proteins in three categories. (**A**) Biological process; (**B**) cellular component; (**C**) molecular function.

**Figure 5 pathogens-14-01068-f005:**
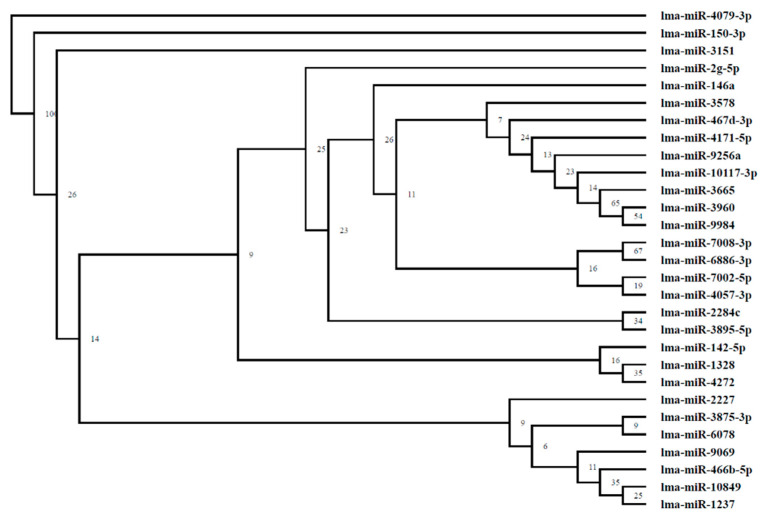
Phylogenetic tree of identified miRNAs in *L. major*.

**Figure 6 pathogens-14-01068-f006:**
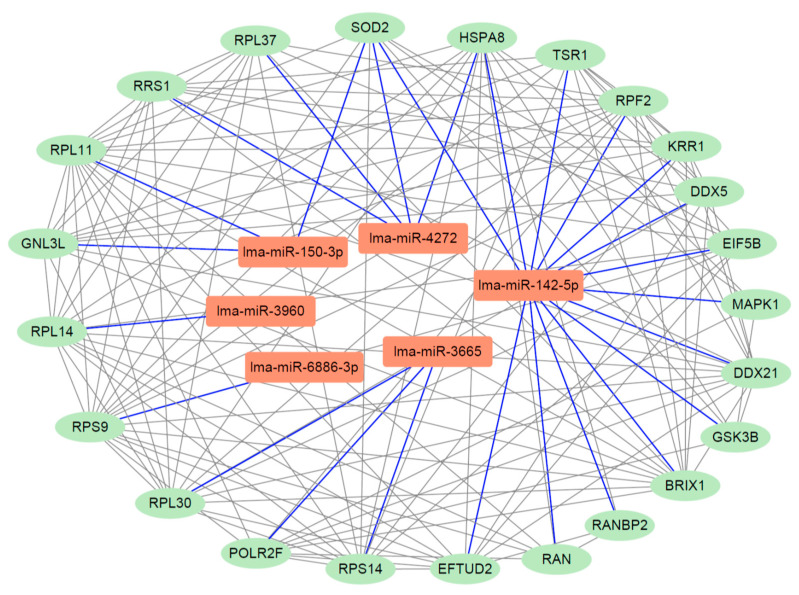
The miRNA-gene interaction in *L. major*. The green node indicated genes, while the red node indicated miRNAs.

**Table 1 pathogens-14-01068-t001:** The annotation of hypothetical proteins with extremely low E-values in *L. major*. The current annotation was retrieved from NCBI database in February 2025.

No.	Accession Number	Current Annotation	Our Annotated Protein	Gene	E-Value
1	XP_001686672.1	Conserved hypothetical protein	Intraflagellar transport protein 122	IFT122	2.27 × 10^−149^
2	XP_888594.1	Conserved hypothetical protein	Tetratricopeptide repeat domain 21B	TTC21B	1.57 × 10^−117^
3	XP_001684649.1	Conserved hypothetical protein	Bardet-Biedl syndrome 2 protein homolog	Bbs2	2.43 × 10^−103^
4	XP_001686229.1	Conserved hypothetical protein	BCS1-like protein	BCS1L	1.14 × 10^−83^
5	XP_001685262.1	Conserved hypothetical protein	Endoplasmic reticulum-Golgi intermediate compartment protein 3	ERGIC3	1.49 × 10^−77^
6	XP_001686195.1	Conserved hypothetical protein	Carnosine N-methyltransferase	CARNMT1	7.04 × 10^−72^
7	XP_001682610.1	Conserved hypothetical protein	Intraflagellar transport 80	IFT80	2.75 × 10^−68^
8	XP_003722084.1	Conserved hypothetical protein	WD repeat domain 34	DYNC2I2	2.63 × 10^−66^
9	XP_003722065.1	Conserved hypothetical protein	Cilia and flagella associated protein 58	CFAP58	5.87 × 10^−61^
10	XP_001682381.1	Conserved hypothetical protein	Tetratricopeptide repeat protein 27	TTC27	1.01 × 10^−58^
11	XP_001684329.1	Conserved hypothetical protein	Zinc finger MYND domain-containing protein 10	ZMYND10	3.96 × 10^−56^
12	XP_001680824.1	Conserved hypothetical protein	Basic immunoglobulin-like variable motif-containing protein	BIVM	5.13 × 10^−56^
13	XP_001683666.1	Conserved hypothetical protein	PWP1 homolog, endonuclein	PWP1	8.23 × 10^−54^
14	XP_003722868.1	Conserved hypothetical protein	Hypoxia up-regulated protein 1	HYOU1	9.27 × 10^−53^
15	XP_001682917.1	Conserved hypothetical protein	tRNA (uracil(54)-C(5))-methyltransferase	TRMT2B	6.68 × 10^−52^

**Table 2 pathogens-14-01068-t002:** Identified novel infection-related proteins in *L. major*.

Group	Accession	Gene	Protein Detail
Kinase	XP_001683373.1	PRKAB2	5′-AMP-activated protein kinase subunit beta-2
Kinase	XP_003722471.1	PRKAG1	5′-AMP-activated protein kinase subunit gamma-1
Kinase	ABE47512.1	ADK2	Adenylate kinase 2
Kinase	XP_001681879.1	ADK7	Adenylate kinase 7
Kinase	XP_001682080.1	ADK8	Adenylate kinase 8
Kinase	XP_001687191.1	ADK9	Adenylate kinase 9
Kinase	XP_001686496.1	BCKDK	Branched-chain alpha-ketoacid dehydrogenase kinase
Kinase	XP_001684075.1	CERK	Ceramide kinase
Kinase	XP_003722687.1	DOLK	Dolichol kinase
Kinase	XP_001687055.1	EEF2K	Eukaryotic elongation factor 2 kinase
Kinase	XP_001684391.1	IDNK	Gluconate kinase
Kinase	XP_001681095.1	HUNK	Hormonally up-regulated neu tumor-associated kinase
Kinase	XP_001684669.1	LRRK2	Leucine-rich repeat serine/threonine-protein kinase 2
Kinase	XP_001682126.1	FCSK	L-fucose kinase
Kinase	QED45586.1	MEK1	MAPK/ERK kinase 1
Kinase	XP_001685169.2	MAP3K1	Mitogen-activated proteinkinase 1
Kinase	XP_003721529.1	CMPK2	Nucleoside-diphosphate kinase
Kinase	XP_001684788.1	PI4KB	Phosphatidylinositol 4-kinase beta
Kinase	AAK28278.1	PGK1	Phosphoglycerate kinase 1
Kinase	XP_001685063.1	PRKCB	Protein kinase C beta type
Kinase	XP_001681082.1	STK16	Serine/threonine-protein kinase 16
Kinase	XP_001685938.1	VPRBP	Serine/threonine-protein kinase VPRBP
Kinase	XP_001683073.1	TK1	Thymidine kinase,
Kinase	XP_001682584.1	YES1	Tyrosine-protein kinase Yes
Tyrosine phosphatase	XP_001681298.1	DUSP12	Dual specificity protein phosphatase 12
Tyrosine phosphatase	XP_001686824.1	ACP3	Protein tyrosine phosphatase ACP3
Tyrosine phosphatase	XP_001683775.1	PTPMT1	Protein-tyrosine phosphatase mitochondrial 1

**Table 3 pathogens-14-01068-t003:** Identified novel miRNAs in *L. major*. Most of the sequences are similar to known miRNAs.

LMA miRNA	miRNA Sequence	Length	EST Accession	Mismatch	MFE	Frequency of MFE
lma-miR-3151	ACGGGTGGCGCAATGGGATCAG	22	AQ901763.1	2	−43.0	3.3%
lma-miR-3665	AGCAGGTGCGGGGCGGCG	18	BH018980.1	1	−40.0	4.9%
lma-miR-3960	GGCGGCGGCGGAGGCGGGGG	20	AQ849868.1	1	−38.2	3.7%
lma-miR-9069	CACGGTGCTGCTGTGGACACG	21	AQ902536.1	2	−36.5	1.9%
lma-miR-3895-5p	ATTGAGTTGATTACGATGG	19	AQ848942.1	2	−35.3	4.3%
lma-miR-146a	TGAGAACTGAATTCCATGGGTT	22	AQ848800.1	0	−35.0	1.6%
lma-miR-150-3p	CTGGTACAGAGGATGGAAGGG	21	AQ901862.1	2	−33.6	2.5%
lma-miR-1328	GAGAGAGAAATGAGAACT	18	BH019430.1	1	−31.6	2.2%
lma-miR-1237	TCTTTCTGCTCCGTCCCCCAG	21	T93471.1	2	−30.4	5.0%
lma-miR-9984	CGCCGCGGCGGCGGCGGC	18	AQ848798.1	0	−30.3	3.2%
lma-miR-7002-5p	TTGGCTTCGGGGAGTACGTGG	21	AL160967.1	2	−30.3	0.7%
lma-miR-10117-3p	GCGGACCATTCAAGATCATC	20	AL354177.1	2	−29.0	4.2%
lma-miR-2284c	AAAAAGTTCGTTTTGGTTTT	20	AQ846415.1	1	−28.1	4.8%
lma-miR-4272	CATTCAACTAGTGATTGT	18	AQ853128.1	2	−27.9	3.3%
lma-miR-3875-3p	TATTTGCGCTAGATAGCGC	19	BH020336.1	2	−27.1	3.6%
lma-miR-2227	TGGCAGTGTTGAAAGACGTC	20	BH885692.1	2	−26.7	2.7%
lma-miR-142-5p	CCCATAAAGTAGAAAGCACT	20	BH018364.1	2	−26.7	4.1%
lma-miR-9256a	TGATCTGGCCACTCAGTGTG	20	BH885819.1	2	−26.2	0.6%
lma-miR-466b-5p	TATGTGTGTGTGTATGTCCATG	22	AA680932.1	1	−25.7	1.4%
lma-miR-6886-3p	TGCCCTTCTCTCCTCCTGCCT	21	AQ847748.1	2	−21.3	3.3%
lma-miR-3578	GAATCCACCACGAACAACTTC	21	AQ850014.1	1	−20.8	1.0%
lma-miR-7008-3p	TGTGCTTCTTGCCTCTTCTCAG	22	AL160615.1	2	−19.1	0.7%
lma-miR-2g-5p	CTCTCCCAATTGTTGTCATGTG	22	AL160715.1	2	−17.9	0.8%
lma-miR-10849	ACTGTGAGCGCAGAATCTCCT	21	AQ845711.1	1	−17.4	0.6%
lma-miR-6078	CCGCCTGAGCTAGCTGTGG	19	AQ843843.1	2	−17.3	2.0%
lma-miR-4171-5p	TGACTCTCTTAAGGAAGCCA	20	BH885526.1	1	−16.7	3.9%
lma-miR-4079-3p	TAGCTCTAGCTGATGTAGCA	20	AQ844078.1	2	−16.4	0.4%
lma-miR-4057-3p	TTTGCTACGGCCACCAAGATCT	22	AQ849400.1	2	−16.2	1.5%
lma-miR-467d-3p	ATATACATACACACACCTACAC	22	AA728206.1	2	−15.4	2.2%

**Table 4 pathogens-14-01068-t004:** Homolog distribution of identified miRNA when compared to other species. The block “YES” indicated that the homolog of corresponding miRNA was present in the corresponding species. Abbreviation of the species: hsa: *Homo sapiens*; mmu: *Mus musculus*; efu: *Eptesicus fuscus*; bta: *Bos taurus*; ptr: *Pan troglodytes*; oan: *Ornithorhynchus anatinus*; eca: *Equus caballus*; cja: *Callithrix jacchus*; oni: *Oreochromis niloticus*; bma: *Brugia malayi*; pte: *Parasteatoda tepidariorum*; cin: *Ciona intestinalis*; hpo: *Heligmosomoides polygyrus*; cbr: *Caenorhabditis briggsae*.

LMA miRNA	Mammal								Non-Mammal					
	hsa	mmu	efu	bta	ptr	oan	eca	cja	oni	bma	pte	cin	hpo	cbr
lma-miR-3151					YES									
lma-miR-3665	YES													
lma-miR-3960	YES	YES												
lma-miR-9069							YES							
lma-miR-3895-5p														
lma-miR-150-3p	YES	YES												
lma-miR-1328						YES								
lma-miR-1237					YES									
lma-miR-9984								YES						
lma-miR-7002-5p		YES												
lma-miR-10117-3p													YES	
lma-miR-2284c				YES										
lma-miR-4272	YES													
lma-miR-3875-3p														
lma-miR-2227														YES
lma-miR-142-5p				YES		YES	YES							
lma-miR-9256a			YES											
lma-miR-466b-5p		YES												
lma-miR-6886-3p	YES													
lma-miR-3578				YES										
lma-miR-7008-3p		YES												
lma-miR-2g-5p										YES	YES			
lma-miR-10849									YES					
lma-miR-6078	YES													
lma-miR-4171-5p												YES		
lma-miR-4079-3p												YES		
lma-miR-4057-3p												YES		
lma-miR-467d-3p		YES												
lma-miR-146a				YES	YES		YES	YES						

## Data Availability

The original contributions presented in this study are included in the article/Appendix A. Further inquiries can be directed to the corresponding author.

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
