# Peer review of "Genome-Wide Identification of Novel miRNAs and Infection-Related Proteins in *Leishmania major* via Comparative Analysis of the Protozoa, Vectors, and Mammalian Hosts"

_pathogens, 2025, doi:10.3390/pathogens14101068_

Round 1
Reviewer 1 Report
Comments and Suggestions for Authors
In this study, authors describe a robust and systematic analysis of miRNAs and protein-coding genes in L. major, its insect vector, and mammalian hosts. The study is interesting and relevant, and some minor points should be considered.
1) In Discussion, the first paragraph is repetitive for what was previously mentioned in the Introduction. This referee recommends that this paragraph be removed.
2) Please species names must be in italics; some “L. major”are not in italics in the manuscript.
Author Response
In this study, authors describe a robust and systematic analysis of miRNAs and protein-coding genes in L. major, its insect vector, and mammalian hosts. The study is interesting and relevant, and some minor points should be considered.
1) In Discussion, the first paragraph is repetitive for what was previously mentioned in the Introduction. This referee recommends that this paragraph be removed.
Reply: Thank you for your suggestion. We have deleted the first paragraph in this new version.
2) Please species names must be in italics; some “L. major” are not in italics in the manuscript.
Reply: Thank you for your suggestion. We have revised all Latin name of species to italics style.

Reviewer 2 Report
Comments and Suggestions for Authors The manuscript presents a comprehensive bioinformatics-based study investigating protein-coding genes and microRNAs in L. major, its sand fly vector, and mammalian hosts. This integrative approach, combining comparative genomics, functional annotation, and network analysis, provides novel insights into parasite pathogenicity and identifies potential molecular targets for therapeutic development. This study offers a significant contribution to our understanding of L. major pathogenicity. The identification of parasite-specific miRNAs and infection-related kinases represents a valuable foundation for biomarker discovery and drug development. The integrated computational approach is rigorous, though experimental validation remains essential to substantiate the therapeutic potential of the predicted targets. With refinement of epidemiological framing and deeper consideration of host-parasite specificity, this manuscript has strong potential for publication and impact in the field of molecular parasitology and drug discovery. Minor points Epidemiological framing: The manuscript states that leishmaniasis accounts for ~17% of infectious diseases worldwide and causes more than 700,000 deaths annually. This figure appears to be an overestimate compared to WHO data, which typically cite an annual incidence of 700,000–1 million new cases and a mortality burden of approximately 20,000–30,000 deaths per year. Updating this information would strengthen accuracy and credibility. Experimental validation: While the computational findings are compelling, their ultimate impact will depend on experimental confirmation. For example, functional assays of the parasite-specific miRNAs, kinase inhibition studies, and host-cell infection models should be outlined as clear next steps to validate predictions. Specificity of targets: The discussion rightly highlights protein kinases and miRNAs as potential therapeutic targets, but additional emphasis should be given to the risk of host cross-reactivity, particularly for conserved kinases. Clarifying how parasite-specific features (sequence motifs, structural differences, expression profiles) could be leveraged to avoid off-target effects would enhance translational relevance. Vector perspective: While the manuscript briefly includes P. papatasi in comparative analyses, its role is underexplored in the discussion. Given the importance of vector-parasite interactions in transmission, future work could better integrate vector-specific findings into the therapeutic or diagnostic framework.Author Response
The manuscript presents a comprehensive bioinformatics-based study investigating protein-coding genes and microRNAs in L. major, its sand fly vector, and mammalian hosts. This integrative approach, combining comparative genomics, functional annotation, and network analysis, provides novel insights into parasite pathogenicity and identifies potential molecular targets for therapeutic development. This study offers a significant contribution to our understanding of L. major pathogenicity. The identification of parasite-specific miRNAs and infection-related kinases represents a valuable foundation for biomarker discovery and drug development. The integrated computational approach is rigorous, though experimental validation remains essential to substantiate the therapeutic potential of the predicted targets. With refinement of epidemiological framing and deeper consideration of host-parasite specificity, this manuscript has strong potential for publication and impact in the field of molecular parasitology and drug discovery.
Reply: We sincerely thank for your thoughtful and encouraging evaluation of our work. We are pleased that the reviewer recognized the integrative nature and translational potential of our bioinformatics-based approach. In response to your valuable suggestions, we have refined the epidemiological context using updated WHO data, expanded the Discussion to address host-parasite specificity and vector considerations. We believe these revisions have strengthened the scientific accuracy and broader relevance of our manuscript.
Minor points
Epidemiological framing: The manuscript states that leishmaniasis accounts for ~17% of infectious diseases worldwide and causes more than 700,000 deaths annually. This figure appears to be an overestimate compared to WHO data, which typically cite an annual incidence of 700,000–1 million new cases and a mortality burden of approximately 20,000–30,000 deaths per year. Updating this information would strengthen accuracy and credibility.
Reply: Thank you for pointing this out. We have checked authoritative sources and corrected the manuscript. According to WHO and related organizations, leishmaniasis causes an estimated 700,000–1,000,000 new cases annually, with ~20,000–30,000 deaths per year. The figure “~17%” and the statement “>700,000 deaths annually” appeared in WHO material referring to vector-borne diseases as a whole, and should not have been attributed to leishmaniasis alone. We have corrected the text and added the following sentence in this updated version.
“According to World Health Organization (WHO), leishmaniasis affects populations in endemic areas, with an estimated 700,000–1,000,000 new cases and approximately 20,000–30,000 deaths per year worldwide [30].”
Experimental validation: While the computational findings are compelling, their ultimate impact will depend on experimental confirmation. For example, functional assays of the parasite-specific miRNAs, kinase inhibition studies, and host-cell infection models should be outlined as clear next steps to validate predictions.
Reply: We thank the reviewer for this valuable suggestion emphasizing the importance of experimental confirmation. In response, we have added the following paragraph in the Discussion section to outline future experimental validation strategies.
“To experimentally validate our computational predictions, several complementary strategies can be employed. First, the parasite-specific miRNAs identified in this study could be functionally characterized using gain- and loss-of-function assays, followed by quantitative assessment of parasite growth and infectivity. The downstream target genes predicted from the miRNA-gene interaction network could be verified through dual-luciferase reporter assays and RT-PCR analyses to confirm regulatory relation-ships [37]. Second, kinase inhibition studies using specific small-molecule inhibitors could help determine the functional importance of the identified infection-related ki-nases in protozoan viability and virulence. Third, macrophage infection models could be applied to evaluate how modulation of these miRNAs or kinase genes affects host–parasite interactions, cytokine responses, and intracellular parasite survival. Together, these approaches will provide direct evidence supporting the bioinformatic predictions and strengthen the mechanistic understanding of L. major virulence.”
Specificity of targets: The discussion rightly highlights protein kinases and miRNAs as potential therapeutic targets, but additional emphasis should be given to the risk of host cross-reactivity, particularly for conserved kinases. Clarifying how parasite-specific features (sequence motifs, structural differences, expression profiles) could be leveraged to avoid off-target effects would enhance translational relevance.
Reply: We appreciate your valuable comment regarding target specificity and the risk of host cross-reactivity. In response, we have added the following paragraph in the Discussion highlighting how parasite-specific sequence motifs, divergent domain architectures, and stage-specific expression profiles can be leveraged to minimize off-target effects. These revisions clarify the translational relevance of our predicted targets.
“While protein kinases and miRNAs emerged as promising therapeutic candidates, the issue of target specificity warrants careful consideration. Because many eukaryotic kinases are evolutionarily conserved [31], there is a potential risk of host cross-reactivity. However, several of the kinases identified here show Leishmania-specific sequence motifs and domain architectures distinct from their mammalian counterparts. For instance, multiple kinase domains display insertions, deletions, or divergent activation-loop residues unique to trypanosomatids, suggesting that selective inhibition may be achievable. Future structural modeling and comparative docking analyses could help pinpoint parasite-specific pockets that differ from those in host homologs. Similarly, transcriptomic and proteomic profiling during infection can further verify parasite-specific expression, minimizing off-target effects on host cells. Collectively, integrating sequence, structural, and expression data will be critical to prioritize Leishmania-specific targets and enhance the translational relevance of these findings.”
Vector perspective: While the manuscript briefly includes P. papatasi in comparative analyses, its role is underexplored in the discussion. Given the importance of vector-parasite interactions in transmission, future work could better integrate vector-specific findings into the therapeutic or diagnostic framework.
Reply: We appreciate your insightful suggestion. We have expanded the Discussion section to better address the vector perspective. The following paragraph has been added to the Discussion.
“In addition to parasite-derived molecular targets, vector–parasite interactions represent a critical but often underexplored aspect of leishmaniasis transmission. P. papatasi, the principal vector of L. major, exhibits species-specific molecular determinants such as midgut receptors and salivary proteins that facilitate parasite attachment and survival [36]. Comparative analyses of vector transcriptomes and proteomes have revealed unique signatures that could serve as biomarkers for potential targets to interrupt transmission. Integrating these vector-specific findings into therapeutic and diagnostic frameworks, such as designing inhibitors that disrupt parasite-vector binding, may enhance the translational potential of Leishmania research. Future work should thus aim to combine parasite and vector omics data to better understand and control leishmaniasis at the molecular interface between parasite and vector.”

Reviewer 3 Report
Comments and Suggestions for Authors
This manuscript describes conserved proteins shared among L. major, vector insects (of several pathogens) and mammalians. Text is well presented, but some adaptations are necessary in order to increase the value of this research for potential readers.
Title – rewrite to read as: Genome-Wide Identification of Novel miRNA and Infection Related Proteins in Leishmania major via Comparative Analysis of THE PROTOZOA, VECTORS, and MAMMALIANS
Consider replacing miRNAs with miRNA – abbreviations do not have a plural form
Always present Latin names in italics
Line 19 – better to call L. major “protozoan”, because insects are also parasites – change accordingly throughout the manuscript
Line 27 – adapt to read as: … NEMATODE Heligosmoides polygyrus
Line 29 – replace pathogenicity with virulence – change accordingly throughout the manuscript
Keywords – display alphabetically
Line 52 – adapt as: BACTERIUM Mycobacterium tuberculosis [6] and FUNGUS Alternaria alternata [7].
Line 60 – adapt to read as: … in species including PLANT Jatropha curcas [9] and PROTOZOAN Trypanosoma 60 brucei [10].
Line 64 – for cutaneous and visceral LEISHMANIASES
Line 89 – provide common names for the mammalian species
Line 94 – Similarly, 10 insect species
Results – present percentages with one decimal place only
Line 193, etc. – species names can be abbreviated after their first use in the abstract and main text, e.g. O. cuniculus (instead of Oryctolagus cuniculus)
Line 216 – 10 insects
Line 231 – and HUMANS
Line 412 – THE leishmaniases…
Line 413 – please check this figure (700,000) and also the cited reference
References – present title in lowercase, as much as possible; present Latin names in italics; journal names should be stadardized
Author Response
This manuscript describes conserved proteins shared among L. major, vector insects (of several pathogens) and mammalians. Text is well presented, but some adaptations are necessary in order to increase the value of this research for potential readers.
Title – rewrite to read as: Genome-Wide Identification of Novel miRNA and Infection Related Proteins in Leishmania major via Comparative Analysis of THE PROTOZOA, VECTORS, and MAMMALIANS
Reply: Thank you for your suggestion. We have revised the title according to your suggestions.
Consider replacing miRNAs with miRNA – abbreviations do not have a plural form
Reply: Thank you for your suggestion. In this manuscript, the word “miRNA” is the abbreviation of “microRNA”, while “miRNAs” is the abbreviation of “microRNAs”. Thus, this is not plural form.
Always present Latin names in italics
Reply: Thank you for your suggestion. We have checked the manuscript, to make all Latin name to italics style.
Line 19 – better to call L. major “protozoan”, because insects are also parasites – change accordingly throughout the manuscript
Reply: Thank you for your good suggestion. We revised all this description of “parasite” to “protozoan” in the manuscript.
Line 27 – adapt to read as: … NEMATODE Heligosmoides polygyrus
Reply: Thank you for your good suggestion. We added “nematode” in the manuscript.
Line 29 – replace pathogenicity with virulence – change accordingly throughout the manuscript
Reply: Thank you for your good suggestion. We have revised all “pathogenicity” to “virulence” in this manuscript.
Keywords – display alphabetically
Reply: Thank you for your good suggestion. We have re-order the keywords alphabetically.
Line 52 – adapt as: BACTERIUM Mycobacterium tuberculosis [6] and FUNGUS Alternaria alternata [7].
Reply: Thank you for your good suggestion. We have added these two words in the sentence.
Line 60 – adapt to read as: … in species including PLANT Jatropha curcas [9] and PROTOZOAN Trypanosoma 60 brucei [10].
Reply: Thank you for your good suggestion. We have added these two words of species classification in the sentence.
Line 64 – for cutaneous and visceral LEISHMANIASES
Reply: Thank you for your good suggestion. We have revised to “leishmaniases”.
Line 89 – provide common names for the mammalian species
Reply: Thank you for your good suggestion. We have added common name for these mammalian species.
Line 94 – Similarly, 10 insect species
Reply: Thank you for your good suggestion. We have added common name for these insect species.
Results – present percentages with one decimal place only
Reply: Thank you for your suggestion. For percentages in Figure 1, because all the percentage values are very close (around 40%), thus we used two decimal places to better discriminate them. For percentage in Table 3, we have revised to one decimal place.
Line 193, etc. – species names can be abbreviated after their first use in the abstract and main text, e.g. O. cuniculus (instead of Oryctolagus cuniculus)
Reply: Thank you for your suggestion. We have revised the Latin name to the abbreviated name in the abstract and main text after their first use.
Line 216 – 10 insects
Reply: Thank you for your suggestion. We have revise to “8 mammals and 10 insects”
Line 231 – and HUMANS
Reply: Thank you for your suggestion. We have revise to “humans”
Line 412 – THE leishmaniases…
Reply: Thank you for your suggestion. According to other reviewer’s comments, the first paragraph of Discussion should be deleted, thus this sentence was deleted in this version.
Line 413 – please check this figure (700,000) and also the cited reference
Reply: Thank you for your suggestion. This comment was also pointed out by other reviewer. We have checked the number, cited reference and added the following sentences in the new version.
“According to World Health Organization (WHO), leishmaniasis affects populations in endemic areas, with an estimated 700,000-1,000,000 new cases and approximately 20,000–30,000 deaths per year worldwide [30].”
References – present title in lowercase, as much as possible; present Latin names in italics; journal names should be standardized
Reply: Thank you for your suggestion. We have revised the reference according to your suggestion.

Round 2
Reviewer 2 Report
Comments and Suggestions for Authors
The authors have adequately answered all the questions raised and the manuscript has been well updated.